# Understanding new tasks through the lens of training data via exponential tilting

**Subha Maity**
Department of Statistics
University of Michigan
smaity@umich.edu

**Mikhail Yurochkin**
IBM Research
MIT-IBM Watson AI lab
mikhail.yurochkin@ibm.com

**Moulinath Banerjee**
Department of Statistics
University of Michigan
moulib@umich.edu

**Yuekai Sun**
Department of Statistics
University of Michigan
yuekai@umich.edu

## ABSTRACT

Deploying machine learning models on new tasks is a major challenge due to differences in distributions of the train (source) data and the new (target) data. However, the training data likely captures some of the properties of the new task. We consider the problem of reweighing the training samples to gain insights into the distribution of the target task. Specifically, we formulate a distribution shift model based on the exponential tilt assumption and learn train data importance weights minimizing the KL divergence between labeled train and unlabeled target datasets. The learned train data weights can then be used for downstream tasks such as target performance evaluation, fine-tuning, and model selection. We demonstrate the efficacy of our method on WATERBIRDS and BREEDS benchmarks. [1]

## 1 INTRODUCTION

Machine learning models are often deployed in a target domain that differs from the domain in which they were trained and validated in. This leads to the practical challenges of adapting and evaluating the performance of models on a new domain without costly labeling of the dataset of interest. For example, in the Inclusive Images challenge (Shankar et al., 2017), the training data largely consists of images from countries in North America and Western Europe. If a model trained on this data is presented with images from countries in Africa and Asia, then (i) it is likely to perform poorly, and (ii) its performance in the training (source) domain may not mirror its performance in the target domain. However, due to the presence of a small fraction of images from Africa and Asia in the source data, it may be possible to reweigh the source samples to mimic the target domain.

In this paper, we consider the problem of learning a set of importance weights so that the reweighted source samples closely mimic the distribution of the target domain. We pose an exponential tilt model of the distribution shift between the train and the target data and an accompanying method that leverages unlabeled target data to fit the model. Although similar methods are widely used in statistics Rosenbaum & Rubin (1983) and machine learning Sugiyama et al. (2012) to train and evaluate models *under covariate shift* (where the decision function/boundary does not change), one of the main benefits of our approach is it allows *concept drift* (where the decision boundary/function are expected to differ) (Cai & Wei, 2019; Gama et al., 2014) between the source and the target domains. We summarize our contributions below:

- In Section 3 we develop a model and an accompanying method for learning source importance weights to mimic the distribution of the target domain *without* labeled target samples.
- In Section 4 we establish theoretical guarantees on the quality of the weight estimates and their utility in the downstream tasks of fine-tuning and model selection.

---

[1] Codes can be found in https://github.com/smaityumich/exponential-tilting.

- We demonstrate applications of our method on WATERBIRDS (Sagawa et al., 2019) (Section 5), BREEDS (Santurkar et al., 2020) (Section 6) and synthetic (Appendix C) datasets.

## 2   RELATED WORK

**Out-of-distribution generalization** is essential for safe deployment of ML models. There are two prevalent problem settings: domain generalization and subpopulation shift (Koh et al., 2020). Domain generalization typically assumes access to several datasets during training that are related to the same task, but differ in their domain or environment (Blanchard et al., 2011; Muandet et al., 2013). The goal is to learn a predictor that can generalize to unseen related datasets via learning invariant representations (Ganin et al., 2016; Sun & Saenko, 2016), invariant risk minimization (Arjovsky et al., 2019; Krueger et al., 2021), or meta-learning (Dou et al., 2019). Domain generalization is a very challenging problem and recent benchmark studies demonstrate that corresponding methods rarely improve over vanilla empirical risk minimization (ERM) on the source data unless given access to labeled target data for model selection (Gulrajani & Lopez-Paz, 2020; Koh et al., 2020).

Subpopulation shift setting assumes that both train and test data consist of the same groups with different group fractions. This setting is typically approached via distributionally robust optimization (DRO) to maximize worst group performance (Duchi et al., 2016; Sagawa et al., 2019), various reweighing strategies (Shimodaira, 2000; Byrd & Lipton, 2019; Sagawa et al., 2020; Idrissi et al., 2021). These methods require group annotations which could be expensive to obtain in practice. Several methods were proposed to sidestep this limitation, however they still rely on a validation set with group annotations for model selection to obtain good performance (Hashimoto et al., 2018; Liu et al., 2021; Zhai et al., 2021; Creager et al., 2021). Our method is most appropriate for the subpopulation shift setting (see Section 3), however it differs in that it does not require group annotations, but requires unlabeled target data.

**Model selection** on out-of-distribution (OOD) data is an important and challenging problem as noted by several authors (Gulrajani & Lopez-Paz, 2020; Koh et al., 2020; Zhai et al., 2021; Creager et al., 2021). Xu & Tibshirani (2022); Chen et al. (2021b) propose solutions specific to covariate shift based on parametric bootstrap and reweighing; Garg et al. (2022); Guillory et al. (2021); Yu et al. (2022) align model confidence and accuracy with a threshold; Jiang et al. (2021); Chen et al. (2021a) train several models and use their ensembles or disagreement. Our importance weighting approach is computationally simpler than the latter and is more flexible in comparison to the former, as it allows for concept drift and can be used in downstream tasks beyond model selection as we demonstrate both theoretically and empirically.

**Domain adaptation** is another closely related problem setting. Domain adaptation (DA) methods require access to labeled source and unlabeled target domains during training and aim to improve target performance via a combination of distribution matching (Ganin et al., 2016; Sun & Saenko, 2016; Shen et al., 2018), self-training (Shu et al., 2018; Kumar et al., 2020), data augmentation (Cai et al., 2021; Ruan et al., 2021), and other regularizers. DA methods are typically challenging to train and require retraining for every new target domain. On the other hand, our importance weights are easy to learn for a new domain allowing for efficient fine-tuning, similar to test-time adaptation methods (Sun et al., 2020; Wang et al., 2020; Zhang et al., 2020), which adjust the model based on the target unlabeled samples. Our importance weights can also be used to define additional regularizers to enhance existing DA methods.

**Importance weighting** has often been used in the domain adaptation literature on label shift (Lipton et al., 2018; Azizzadenesheli et al., 2019; Maity et al., 2022) and covariate shift (Sugiyama et al., 2007; Hashemi & Karimi, 2018) but the application has been lacking in the area of concept drift models (Cai & Wei, 2019; Maity et al., 2021), due to the reason that it is generally impossible to estimate the weights without seeing labeled data from the target. In this paper, we introduce an exponential tilt model which accommodates concept drift while allowing us to estimate the importance weights for the distribution shift.

## 3   THE EXPONENTIAL TILT MODEL

**Notation**  We consider a $K$-class classification problem. Let $\mathcal{X} \in \mathbf{R}^d$ and $\mathcal{Y} \triangleq [K]$ be the space of inputs and set of possible labels, and $P$ and $Q$ be probability distributions on $\mathcal{X} \times \mathcal{Y}$ for the

source and target domains correspondingly. A (probabilistic) classifier is a map $f : \mathcal{X} \to \Delta^{K-1}$. We define $p\{x, Y = k\}$ as the weighted source class conditional density, *i.e.* $p\{x, Y = k\} = p\{x \mid Y = k\} \times P\{Y = k\}$, where $p\{x \mid Y = k\}$ is the density of the source feature distribution in class $k$ and $P\{Y = k\}$ is the class probability in source. We similarly define $q\{x, Y = k\}$ for target.

We consider the problem of learning importance weights on samples from a source domain so that the weighted source samples mimic the target distribution. We assume that the learner has access to labeled samples $\{(X_{P,i}, Y_{P,i})\}_{i=1}^{n_P}$ from the source domain and and unlabeled samples $\{X_{Q,i}\}_{i=1}^{n_Q}$ from the target domain. The learner's goal is to estimate a weight function $\omega(x, y) > 0$ such that

$$\mathbf{E}\left[\omega(X_P, Y_P)g(X_P, Y_P)\right] \approx \mathbf{E}\left[g(X_Q, Y_Q)\right] \text{ for all (reasonable) } g : \mathcal{X} \times \mathcal{Y} \to \mathbf{R}. \tag{3.1}$$

Ideally, $\omega = \frac{dQ}{dP}$ is the likelihood ratio between the source and target domains (this leads to equality in (3.1)), but learning this weight function is generally impossible without labeled samples from the target domain (David et al., 2010). Thus we must impose additional restrictions on the domains.

**The exponential tilt model**  We assume that there is a vector of sufficient statistics $T : \mathcal{X} \to \mathbf{R}^p$ and the parameters $\{\theta_k \in \mathbf{R}^p, \alpha_k \in \mathbf{R}\}_{k=1}^K$ such that

$$\log \frac{q\{x, Y=k\}}{p\{x, Y=k\}} = \theta_k^\top T(x) + \alpha_k \text{ for all } k \in [K]; \tag{3.2}$$

*i.e.* $q\{x, Y = k\}$ is a member of the exponential family with base measure $p\{x, Y = k\}$ and sufficient statistics $T$. We call (3.2) the **exponential tilt** model. It implies the importance weights between the source and target samples are

$$\omega(x, y) = \exp(\theta_y^\top T(x) + \alpha_y).$$

**Model motivation**  The exponential tilt model is motivated by the rich theory of exponential families in statistics. In machine learning, it was used for learning with noisy labels and for improving worst-group performance when group annotations are available (Li et al., 2020; 2021). It is also closely related to several common models in transfer learning and domain adaptation. In particular, it implies there is a linear concept drift between the source and target domains. It also extends the widely used **covariate shift** (Sugiyama & Kawanabe, 2012) and **label shift** models (Alexandari et al., 2020; Lipton et al., 2018; Azizzadenesheli et al., 2019; Maity et al., 2022; Garg et al., 2020) of distribution shifts. It extends the covariate shift model because the exponential tilt model permits (linear in $T(X)$) **concept drifts** between the source and target domains; it extends the label shift model because it allows the class conditionals to differ between the source and target domains. It does, however, come with a limitation: implicit in the model is the assumption that there is some amount of overlap between the source and target domains. In the subpopulation shift setting, this assumption is always satisfied, while in domain generalization it may be violated if the new domain drastically differs from the source data (see Appendix C for a synthetic data example).

**Choosing $T$**  The goal of $T$ is to identify the common subpopulations across domains, such that

$$(X_P, Y_P) \mid \{T(X_P) = t, Y_P = k\} \overset{d}{\approx} (X_Q, Y_Q) \mid \{T(X_Q) = t, Y_P = k\}.$$

If $T$ segments the source domain into its subpopulations (*i.e.* the subpopulations are $\{(x, y) \in \mathcal{X} \times \mathcal{Y} \mid T(x) = t, y = k\}$ for different values of $t$'s and $k$'s), then it is possible to achieve perfect reweighing of the source domain with the exponential tilt model: the weight of the $\{T(X) = t, Y = k\}$ subpopulation is $\exp(\theta_k^\top t + \alpha_k)$. However, in practice, such a $T$ that perfectly segments the subpopulations may not exist (*e.g.* the subpopulations may overlap) or is very hard to learn (*e.g.* we don't have prior knowledge of the subpopulations to guide $T$).

If no prior knowledge of the domains is available, we can use a neural network to parameterize $T$ and learn its weights along with the tilt parameters, or simply use a pre-trained feature extractor as $T$, which we demonstrate to be sufficiently effective in our empirical studies. We also study the effects of misspecification of $T$ using a synthetic dataset example in Appendix C.

**Fitting the exponential tilt model**  We fit the exponential tilt model via distribution matching. This step is based on the observation that under the exponential tilt model (3.2)

$$q_X\{x\} = \sum_{k=1}^K p\{x, Y = k\}\exp(\theta_k^\top T(x) + \alpha_k), \tag{3.3}$$

where $q_X$ is the (marginal) density of the inputs in the target domain. It is possible to obtain an estimate $\widehat{q}_X$ of $q_X$ from the unlabeled samples $\{X_{i,Q}\}_{i=1}^n$ and estimates $\widehat{p}\{x, Y = k\}$ of the $p\{x, Y = k\}$'s from the labeled samples $\{(X_{i,P}, Y_{i,P})\}_{i=1}^m$. This suggests we find $\theta_k$'s and $\alpha_k$'s such that

$$\textstyle\sum_{k=1}^K \widehat{p}\{x, Y = k\}\exp(\theta_k^\top T(x) + \alpha_k) \approx \widehat{q}_X\{x\}.$$

Note that the $\theta_k$'s and $\alpha_k$'s are dependent because $\widehat{q}_X$ must integrate to one. We enforce this restriction as a constraint in the distribution matching problem:

$$\{(\hat{\theta}_k, \hat{\alpha}_k)\}_{k=1}^K \in \begin{cases} \arg\min_{\{(\theta_k, \alpha_k)\}_{k=1}^K} D\left(\widehat{q}_X\{x\} \| \sum_{k=1}^K \widehat{p}\{x, Y = k\}\exp(\theta_k^\top T(x) + \alpha_k)\right) \\ \text{subject to } \int_\mathcal{X} \sum_{k=1}^K \widehat{p}\{x, Y = k\}\exp(\theta_k^\top T(x) + \alpha_k)dx = 1\,, \end{cases}$$
$$(3.4)$$

where $D$ is a discrepancy between probability distributions on $\mathcal{X}$. Although there are many possible choices of $D$, we pick the Kullback-Leibler (KL) divergence in the rest of this paper because it leads to some computational benefits. We reformulate the above optimization for KL-divergence to relax the constraint which we state in the following lemma.

**Lemma 3.1.** *If $D$ is the Kullback-Leibler (KL) divergence then optima in (3.3) is achieved at $\{(\hat{\theta}_k, \hat{\alpha}_k)\}_{k=1}^K$ where*

$$\{(\hat{\theta}_k, \hat{\alpha}'_k)\}_{k=1}^K \in \arg\max_{\{(\theta_k, \alpha'_k)\}_{k=1}^K} \mathbf{E}_{\widehat{Q}_X}\left[\log\left\{\sum_{k=1}^K \widehat{\eta}_{P,k}(X)exp(\theta_k^\top T(X) + \alpha'_k)\right\}\right.$$
$$\left. - \log\left\{\mathbf{E}_{\widehat{P}}\left[exp(\theta_Y^\top T(X) + \alpha'_Y)\right]\right\}\right\}$$

*$\widehat{\eta}_P = \{\widehat{\eta}_{P,k}\}_{k=1}^K$ is a probabilistic classifier for $P$ and $\hat{\alpha}_k = \hat{\alpha}'_k - \log\left\{\mathbf{E}_{\widehat{P}}\left[exp(\hat{\theta}_Y^\top T(X) + \hat{\alpha}'_Y)\right]\right\}$.*

One benefit of minimizing the KL divergence is that the learner does not need to estimate the $p\{x, Y = k\}$'s, a generative model that is difficult to train. They merely need to train a discriminative model to estimate $\widehat{\eta}_P$ from the (labeled) samples from the source domain.

We plug the fitted $\hat{\theta}_k$'s and $\hat{\alpha}_k$'s into (3.5) to obtain Exponential Tilt Reweighting Alignment (ExTRA) importance weights:

$$\widehat{\omega}(x, y) = \exp(\hat{\theta}_y^\top T(x) + \hat{\alpha}_y). \tag{3.5}$$

We summarize the ExTRA procedure in Algorithm 1 in Appendix B.2.

Next we describe two downstream tasks where ExTRA weights can be used:

1. **ExTRA model evaluation in the target domain.** Practitioners may estimate the target performance of a model in the target domain by reweighing the empirical risk in the source domain:

$$\mathbf{E}\left[\ell(f(X_Q), Y_Q)\right] \approx \frac{1}{n_P}\sum_{i=1}^{n_P} \ell(f(X_{P,i}), Y_{P,i})\widehat{\omega}(X_{P,i}, Y_{P,i}), \tag{3.6}$$

   where $\ell$ is a loss function. This allows to evaluate models in the target domain without target labeled samples *even in the presence of concept drift between the training and target domain*.
2. **ExTRA fine-tuning for target domain performance.** Since the reweighted empirical risk (in the source domain) is a good estimate of the risk in the target domain, practitioners may fine-tune models for the target domain by minimizing the reweighted empirical risk:

$$\widehat{f}_Q \in \arg\min_{f \in \mathcal{F}} \mathbf{E}_{\widehat{P}}\left[\ell(f(X), Y)\widehat{\omega}(X, Y)\right]. \tag{3.7}$$

We note that the correctness of (3.4) depends on the identifiability of the $\theta_k$'s and $\alpha_k$'s from (3.3); *i.e.* the uniqueness of the parameters that satisfy (3.3). As long as the tilt parameters are identifiable, then (3.4) provides consistent estimates of them. However, without additional assumptions on the $p\{x, Y = k\}$'s and $T$, the tilt parameters are generally unidentifiable from (3.3). Next we elaborate on the identifiability of the exponential tilt model.

## 4 Theoretical properties of exponential tilting

### 4.1 Identifiability of the exponential tilt model

To show that the $\theta_k$'s and $\alpha_k$'s are identifiable from (3.3), we must show that there is a unique solution to (3.3). Unfortunately, this is not always the case. For example, consider a linear discriminant

analysis (LDA) problem in which the class conditionals drift between the source and target domains:

$$p\{x, Y = k\} = \pi_k \phi(x - \mu_{P,k}), \quad q\{x, Y = k\} \quad = \pi_k \phi(x - \mu_{Q,k}),$$

where $\phi$ is the standard multivariate normal density, $\pi_k \in (0, 1)$ are the class proportions in both source and target domains, and $\mu_{P,k}$'s (resp. the $\mu_{Q,k}$'s) are the class conditional means in the source (resp. target) domains. We see that this problem satisfies the exponential tilt model with $T(x) = x$:

$$\log \frac{q\{x, Y=k\}}{p\{x, Y=k\}} = (\mu_{Q,k} - \mu_{P,k})^\top x - \tfrac{1}{2}\|\mu_{Q,k}\|_2^2 + \tfrac{1}{2}\|\mu_{P,k}\|_2^2.$$

This instance of the exponential tilt model is not identifiable. Any permutation of the class labels $\sigma : [K] \to [K]$ also leads to the same (marginal) distribution of inputs:

$$\sum_{k=1}^K p\{x, Y = k\}\exp\left((\mu_{Q,k} - \mu_{P,k})^\top x + \tfrac{1}{2}\|\mu_{P,k}\|_2^2 - \tfrac{1}{2}\|\mu_{Q,k}\|_2^2\right)$$
$$= \sum_{k=1}^K p\{x, Y = k\}\exp\left((\mu_{Q,\sigma(k)} - \mu_{P,k})^\top x + \tfrac{1}{2}\|\mu_{P,k}\|_2^2 - \tfrac{1}{2}\|\mu_{Q,\sigma(k)}\|_2^2\right).$$

From this example, we see that the non-identifiability of the exponential tilt model is closely related to the label switching problem in clustering. Intuitively, the exponential tilt model in the preceding example is too flexible because it can tilt any $p\{x, Y = k\}$ to $q\{x, Y = l\}$. Thus there is ambiguity in which $p\{x, Y = k\}$ tilts to which $q\{x, Y = l\}$. In the rest of this subsection, we present an identification restriction that guarantees the identifiability of the exponential tilt model.

A standard identification restriction in related work on domain adaptation is a clustering assumption. For example, Tachet et al. (2020) assume there is a partition of $\mathcal{X}$ into disjoint sets $\mathcal{X}_k$ such that $\text{supp}(P\{\cdot \mid Y = k\})$, $\text{supp}(Q\{\cdot \mid Y = k\}) \subset \mathcal{X}_k$ for all $k \in [K]$. This assumption is strong: it implies there is a perfect classifier in the source and target domains. Here we consider a weaker version of the clustering assumption: there are sets $\mathcal{S}_k$ such that

$$P\{Y = k \mid X \in \mathcal{S}_k\} = Q\{Y = k \mid X \in \mathcal{S}_k\} = 1.$$

We note that the $\mathcal{S}_k$'s can be much smaller than the $\mathcal{X}_k$'s; this permits the supports of $P\{\cdot \mid Y = k\}$ and $P\{\cdot \mid Y = l\}$ to overlap.

**Definition 4.1** (anchor set). *A set $\mathcal{S}_k \subset \mathcal{X}$ is an **anchor set** for class $k$ if $p\{x, Y = k\} > 0$ and $p\{x, Y = l\} = 0$, $l \neq k$ for all $x \in \mathcal{S}_k$.*

**Proposition 4.2** (identifiability from anchor sets). *If there are anchor sets $\mathcal{S}_k$ for all $K$ classes (in the source domain) and $T(\mathcal{S}_k)$ is $p$-dimensional, then there is at most one set of $\theta_k$'s and $\alpha_k$'s that satisfies (3.3).*

This identification restriction is also closely related to the linear independence assumption in Gong et al. (2016). Inspecting the proof of proposition 4.2 (see Appendix A.3), we see that the anchor set assumption implies linear independence of $\{p_k(x)\exp(\theta_k^\top T(x) + \alpha_k)\}_{k=1}^K$ for any set of $\theta_k$'s and $\alpha_k$'s. We study the anchor set assumption empirically in a synthetic experiment in Appendix C. Our experiments show that the assumption is mild and is violated only under extreme data scenarios.

## 4.2 CONSISTENCY IN ESTIMATION OF THE TILT PARAMETERS

Here, we establish a convergence rate for the estimated tilt parameters (Lemma (3.1)) and the Ex-TRA importance weights (Equation (3.1)). To simplify the notation, we define $S(x) = (1, T(x)^\top)^\top$ as the extended sufficient statistics for the exponential tilt and denote the corresponding tilt parameters as $\xi_k = (\alpha_k, \theta_k^\top)^\top$. We let $\xi_k^\star = (\alpha_k^\star, \theta_k^{\star\top})^\top$'s be the true values of the tilt parameters $\xi_k$'s and let $\xi = (\xi_1^\top, \dots, \xi_K^\top)^\top \in \mathbf{R}^{K(p+1)}$ be the long vector containing all the tilt parameters. We recall that estimating the parameters from the optimization stated in Lemma 3.1 requires a classifier $\widehat{\eta}_P$ on the source data. So, we define our objective for estimating $\xi$ through a generic classifier $\eta : \mathcal{X} \to \Delta^K$. Denoting $\eta_k(x)$ as the $k$-th co-ordinate of $\eta(x)$ we define the expected log-likelihood objective as:

$$\mathfrak{L}(\eta, \xi) = \mathbf{E}_{Q_X}[\log\{\textstyle\sum_{k=1}^K \eta_k(X)\exp(\xi_k^\top S(X))\}] - \log[\mathbf{E}_P\{\exp(\xi_Y^\top S(X))\}],$$

and its empirical version as

$$\hat{\mathfrak{L}}(\eta, \xi) = \mathbf{E}_{\widehat{Q}_X}[\log\{\textstyle\sum_{k=1}^K \eta_k(X)\exp(\xi_k^\top S(X))\}] - \log[\mathbf{E}_{\widehat{P}}\{\exp(\xi_Y^\top S(X))\}].$$

To establish the consistency of MLE we first make an assumption that the loss $\xi \mapsto -\mathfrak{L}(\eta_P^\star, \xi)$ is strongly convex at the true parameter value.

**Assumption 4.3.** *The loss $\xi \mapsto -\mathfrak{L}(\eta_P^\star, \xi)$ is strongly convex at $\xi^\star$, i.e., there exists a constant $\mu > 0$ such that for any $\xi$ it holds:*

$$-\mathfrak{L}(\eta_P^\star, \xi) \geq -\mathfrak{L}(\eta_P^\star, \xi^\star) - \partial_\xi \mathfrak{L}(\eta_P^\star, \xi^\star)^\top (\xi - \xi^\star) + \frac{\mu}{2} \|\xi - \xi^\star\|_2^2 \,.$$

We note that the assumption is a restriction on the distribution $Q$ rather than the objective itself. For technical convenience we next assume that the feature space is bounded.

**Assumption 4.4.** *$\mathcal{X}$ is bounded, i.e., there exists an $M > 0$ such that $\mathcal{X} \subset B(0, M)$.*

Recall, from Lemma 3.1, that we need a fitted source classifier $\widehat{\eta}_P$ to estimate the tilt parameter: $\xi^\star$ is estimated by maximizing $\hat{\mathfrak{L}}(\widehat{\eta}_P, \xi)$ rather than the unknown $\hat{\mathfrak{L}}(\eta_P^\star, \xi)$. While analyzing the convergence of $\hat{\xi}$ we are required to control the difference $\hat{\mathfrak{L}}(\hat{\eta}_P, \xi) - \hat{\mathfrak{L}}(\eta_P^\star, \xi)$. To ensure the difference is small, assume the pilot estimate of the source regression function $\widehat{\eta}_P$ is consistent at some rate $r_{n_P}$.

**Assumption 4.5.** *Let $f_{P,k}^\star(x) = \log\{\eta_{P,k}^\star(x)\} - \frac{1}{K} \sum_{j=1}^K \log\{\eta_{P,j}^\star(x)\}$. We assume that there exist an estimators $\{\hat{f}_{P,k}(x)\}_{k=1}^K$ for $\{f_{P,k}^\star(x)\}_{k=1}^K$ such that the following holds: there exists a constant $c > 0$ and a sequence $r_{n_P} \to 0$ such that for almost surely $[\mathbb{P}_X]$ it holds*

$$\mathbf{P}(\|\hat{f}_P(x) - f_P^\star(x)\|_2 > t) \leq exp(-ct^2/r_{n_P}^2), \ t > 0 \,.$$

We use the estimated logits $\{\hat{f}_{P,k}(x)\}_{k=1}^K$ to construct the regression functions as $\widehat{\eta}_{P,k}(x) = \exp(\hat{f}_{P,k}(x))/\{\sum_{j=1}^K \exp(\hat{f}_{P,j}(x))\}$, which we use in the objective stated in Lemma 3.1 to analyze the convergence of the tilt parameter estimates and the ExTRA weights. With the above assumptions we're now ready to state concentration bounds for $\hat{\xi} - \xi^\star$ and $\hat{\omega} - \omega^\star$, where the true importance weight $\omega^\star$ is defined as $\omega^\star(x, y) = \exp(\xi_y^{\star\top} S(x))$.

**Theorem 4.6.** *Let the assumptions 4.3, 4.4 and 4.5 hold. For the sample sizes $n_P, n_Q$ define $\alpha_{n_P, n_Q} = r_{n_P} \sqrt{\log(n_Q)} + \{(p+1)K/n_P\}^{1/2} + \{(p+1)K/n_Q\}^{1/2}$. There exists constants $k_1, k_2 > 0$ such that for any $\delta > 0$ with probability at least $1 - (2K + 1)\delta$ the following hold:*

$$\|\hat{\xi} - \xi^\star\|_2 \leq k_1 \alpha_{n_P, n_Q} \sqrt{\log(1/\delta)}, \ and \ \|\hat{\omega} - \omega^\star\|_{1,P} \leq k_2 \alpha_{n_P, n_Q} \sqrt{\log(1/\delta)}.$$

In Theorem 4.6 we notice that as long as $r_{n_P} \log(n_Q) \to 0$ for $n_P, n_Q \to \infty$ we have $\alpha_{n_P, n_Q} \to 0$. This implies both *the estimated tilt parameters and the ExTRA weights converge to their true values as the sample sizes $n_P, n_Q \to \infty$.*

We next provide theoretical guarantees for the downstream tasks (1) fine-tuning and (2) target performance evaluation that we described in Section 3.

**Fine-tuning** We establish a generalization bound for the fitted model (3.7) using weighted-ERM on source domain. We denote $\mathcal{F}$ as the classifier hypothesis class. For $f \in \mathcal{F}$ and a weight function $\omega : \mathcal{X} \times \mathcal{Y} \to \mathbf{R}_{\geq 0}$ define the weighted loss function and its empirical version on source data as:

$$\mathcal{L}_P(f, w) = \mathbf{E}_P[\omega(X, Y)\ell(f(X), Y)], \ \hat{\mathcal{L}}_P(f, w) = \mathbf{E}_{\widehat{P}}[\omega(X, Y)\ell(f(X), Y)] \,.$$

We also define the loss function on the target data as: $\mathcal{L}_Q(f) = \mathbf{E}_Q[\ell(f(X), Y)]$. If $\{(\theta_k^\star, \alpha_k^\star)\}_{k=1}^K$ is the true value of the tilt parameters in (3.2), *i.e.*, the following holds:

$$q\{x, Y = k\} = p\{x, Y = k\}\exp\{\alpha_k^\star + (\theta_k^\star)^\top T(x)\}; \ k \in [K] \,,$$

then defining $\omega^\star(x, k) = \exp\{\alpha_k^\star + (\theta_k^\star)^\top T(x)\}$ as the true weight we notice that $\mathcal{L}_P(f, \omega^\star) = \mathcal{L}_Q(f)$, which is easily observed by setting $g(x, y) = \ell(f(x), y)$ in the display (3.1).

To establish our generalization bound we require Rademacher complexity (Bartlett & Mendelson, 2002) (denoted as $\mathcal{R}_{n_P}(\mathcal{G})$; see Appendix A.1 for details) and the following assumption on the loss function.

**Assumption 4.7.** *The loss function $\ell$ is bounded, i.e., for some $B > 0$, $|\ell\{f(x), y\}| \leq B$ for any $f \in \mathcal{F}$, $x \in \mathcal{X}$ and $y \in [K]$.*

With the above definitions and the assumption we establish our generalization bound.

**Lemma 4.8.** *For a weight function $\omega$ and the source samples $\{(X_{P,i}, Y_{P,i})\}_{i=1}^{n_P}$ of size $n_P$ let $\hat{f}_\omega = \arg\min_{f \in \mathcal{F}} \hat{\mathcal{L}}_P(f, \omega)$. There exists a constant $c > 0$ such that the following generalization bound holds with probability at least $1 - \delta$*

$$\mathcal{L}_Q(\hat{f}_\omega) - \min_{f \in \mathcal{F}} \mathcal{L}_Q(f) \leq 2\mathcal{R}_{n_P}(\mathcal{G}) + B\|\omega - \omega^\star\|_{1,P} + c\sqrt{\frac{\log(1/\delta)}{n_P}}, \qquad (4.1)$$

*where $\mathcal{R}_{n_P}(\mathcal{G})$ is the Rademacher complexity of $\mathcal{G} = \{\omega^\star(x,y)\ell(f(x),y) : f \in \mathcal{F}\}$ defined in Appendix A.1.*

In Theorem 4.6 we established an upper bound for the estimated weights $\hat{\omega}$, which concludes that $\hat{f}_{\hat{\omega}}$ has the following generalization bound: for any $\delta > 0$, with probability at least $1 - (2K + 2)\delta$

$$\mathcal{L}_Q(\hat{f}_{\hat{\omega}}) - \min_{f \in \mathcal{F}} \mathcal{L}_Q(f) \leq 2\mathcal{R}_{n_P}(\mathcal{G}) + k_2 \alpha_{n_P, n_Q}\sqrt{\log(1/\delta)} + c\sqrt{\log(1/\delta)/n_P},$$

where $k_2$ is the constant in Theorem 4.6 and $c$ is the constant in Lemma 4.8. The generalization bound implies that for large sample sizes ($n_P, n_Q \to \infty$) the target accuracy of weighted ERM on source data well approximates the accuracy of ERM on target data.

**Target performance evaluation**  We provide a theoretical guarantee for the target performance evaluation (3.6) using our importance weights. Here we only consider the functions $g : \mathcal{X} \times \mathcal{Y} \to \mathbf{R}$ which are bounded by some $B > 0$, *i.e.* $|g(x,y)| \leq B$ for all $x \in \mathcal{X}$ and $y \in \mathcal{Y}$. The simplest and the most frequently used example is the model accuracy which uses 0-1-loss as the loss function: for a model $f$ the loss $g(x,y) = \mathbb{I}\{f(x) = y\}$ is bounded with $B = 1$. For such functions we notice that $\mathbf{E}_Q[g(X,Y)] = \mathbf{E}_P[g(X,Y)\omega^\star(X,Y)]$, as observed in display (3.1). This implies the following bound on the target performance evaluation error

$$\left|\mathbf{E}_Q[g(X,Y)] - \mathbf{E}_P[g(X,Y)\hat{\omega}(X,Y)]\right| = \left|\mathbf{E}_P[g(X,Y)\omega^\star(X,Y)] - \mathbf{E}_P[g(X,Y)\hat{\omega}(X,Y)]\right|$$
$$\leq B\mathbf{E}_P[|\hat{\omega}^\star(X,Y) - \omega^\star(X,Y)|] \leq B\|\hat{\omega} - \omega^\star\|_{1,P}.$$

We recall the concentration bound for $\|\hat{\omega} - \omega^\star\|_{1,P}$ from Theorem 4.6 and conclude that the *estimated target performance in* (3.6) *converges to the true target performance* at rate $\alpha_{n_P, n_Q}$.

## 5  WATERBIRDS CASE STUDY

To demonstrate the efficacy of the ExTRA algorithm for reweighing the source data we (i) verify the ability of ExTRA to upweigh samples most relevant to the target task; (ii) evaluate the utility of weights in downstream tasks such as fine-tuning and (iii) model selection.

**WATERBIRDS dataset** combines bird photographs from the Caltech-UCSD Birds-200-2011 (CUB) dataset (Wah et al., 2011) and the image backgrounds from the Places dataset (Zhou et al., 2017). The birds are labeled as one of $\mathcal{Y} = \{\text{waterbird, landbird}\}$ and placed on one of $\mathcal{A} = \{\text{water background, land background}\}$. The images are divided into four groups: landbirds on land (0); landbirds on water (1); waterbirds on land (2); waterbirds on water (3). The source dataset is highly imbalanced, i.e. the smallest group (2) has 56 samples. We embed all images with a pre-trained ResNet18 (He et al., 2016). See Appendix B.1 for details.

We consider five subpopulation shift target domains: all pairs of domains with different bird types and the original test set (Sagawa et al., 2019) where all 4 groups are present with proportions vastly different from the source. For all domains, we fit ExTRA weights (with ResNet18 features as $T(x)$) from 10 different initializations and report means and standard deviations for the metrics. See Appendix B.2 for the implementation details.

**ExTRA weights quality** For a given target domain it is most valuable to upweigh the samples in the source data corresponding to the groups comprising that domain. The most challenging is the target $\{1, 2\}$ consisting only of birds appearing on their atypical backgrounds. Groups $\{1, 2\}$ correspond to 5% of the source data making them most difficult to "find". To quantify the ability of ExTRA to upweigh these samples we report precision (proportion of samples from groups $\{1, 2\}$ within the top $x\%$ of the weights) and recall (proportion of $\{1, 2\}$ samples within the top $x\%$ of the weights) in Figure 1. We notice that samples corresponding to 10% largest ExTRA weights contain slightly

Table 1: Model selection results on WATERBIRDS

| target groups | target accuracy | | | rank correlation | | |
|---|---|---|---|---|---|---|
| | ExTRA | SrcVal | ATC-NE | ExTRA | SrcVal | ATC-NE |
| $\{0, 2\}$ | 0.819±0.012 | 0.854 | **0.871** | 0.419±0.01 | **0.807** | 0.760 |
| $\{1, 2\}$ | **0.741**±0.047 | 0.616 | 0.646 | **0.747**±0.106 | -0.519 | -0.590 |
| $\{0, 3\}$ | **0.978**±0.001 | **0.978** | 0.976 | **0.962**±0.004 | 0.956 | 0.906 |
| $\{1, 3\}$ | **0.757**±0.011 | 0.737 | 0.747 | **0.361**±0.168 | -0.318 | -0.411 |
| $\{0, 1, 2, 3\}$ | **0.856**±0.034 | 0.803 | 0.818 | **0.658**±0.295 | 0.263 | 0.178 |
| average | **0.83** | 0.798 | 0.812 | **0.753** | 0.166 | 0.110 |

over 80% of the groups $\{1, 2\}$ in the source data (recall). This demonstrates the ability of ExTRA to upweigh relevant samples. We present examples of upweighted images and results for other target domains in Appendix B.3.

**Model fine-tuning** We demonstrate the utility of ExTRA weights in the fine-tuning (3.7). The basic goal of such importance weighing is to improve the performance in the target in comparison to training on uniform source weights S -> T, *i.e.* ERM. Another baseline is the DRO model (Hashimoto et al., 2018) that aims to maximize worst-group performance without access to the group labels, and JTT (Liu et al., 2021) that retrains a model after upweighting the misclassified samples by ERM. We consider two additional baselines that utilize group annotations to improve worst-group performance: re-weighing the source to equalize group proportions ($\text{RW}_{\text{gr}}$) and group DRO (gDRO) (Sagawa et al., 2019). The afore-

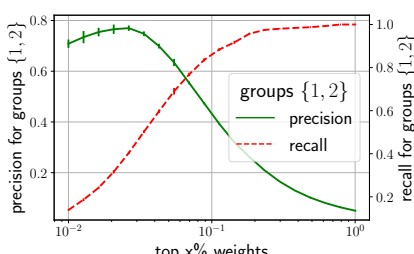

Figure 1: ExTRA precision and recall for samples with top $x\%$ weights.

mentioned baselines do not try to adjust to the target domain. Finally, we compare to $\pi$T -> T that fine-tunes the model only using the subset of the source samples corresponding to the target domain groups. In all cases we use logistic regression as model class.

We compare target accuracy across domains in Figure 2. Analogous comparison with area under the receiver operator curve can be found in Figure 6 in Appendix B.3.1. Model trained with ExTRA weights outperforms all "fair" baselines and matches the performance of the three baselines that had access to additional information. In all target domains ExTRA fine-tuning is comparable with the $\pi$T -> T supporting its ability to upweigh relevant samples. Notably, on $\{1,2\}$ domain of both minority groups *and* on $\{0,3\}$ domain of both majority groups, ExTRA outperforms $\text{RW}_{\text{gr}}$ and gDRO that utilize group annotations. This emphasizes the advantage of adapting to the target domain instead of pursuing a more conser-

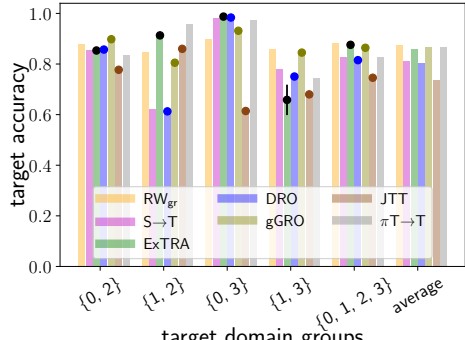

Figure 2: Performance on WATERBIRDS.

vative goal of worst-group performance maximization. Finally, we note that ExTRA fine-tuning did not perform as well on the domain $\{1,3\}$, however neither did $\pi$T -> T.

**Model selection** out-of-distribution is an important task, that is difficult to perform without target data labels and group annotations (Gulrajani & Lopez-Paz, 2020; Zhai et al., 2021). We evaluate the ability of choosing a model for the target domain based on accuracy on the ExTRA reweighted source validation data. We compare to the standard source validation model selection (SrcVal) and to the recently proposed ATC-NE (Garg et al., 2022) that uses negative entropy of the predicted probabilities on the target domain to score models. We fit a total of 120 logistic regression models with different weighting (uniform, label balancing, and group balancing) and varying regularizers. See Appendix B.2 for details.

In Table 1 we compare the target performance of models selected using each of the model evaluation scores and rank correlation between the corresponding model scores and true target accuracies.

Model selection with ExTRA results in the best target performance and rank correlation on 4 out of 5 domains and on average. Importantly, the rank correlation between the true performance and ExTRA model scores is always positive, unlike the baselines, suggesting its reliability in providing meaningful information about the target domain performance.

## 6  BREEDS CASE STUDY

BREEDS (Santurkar et al., 2020) is a subpopulation shift benchmark derived from ImageNet (Deng et al., 2009). It uses the class hierarchy to define groups within classes. For example, in the Entity-30 task considered in this experiment, class fruit is represented by strawberry, pineapple, jackfruit, Granny Smith in the source and buckeye, corn, ear, acorn in the target. This is an extreme case of subpopulation shift where source and target groups have zero overlap. We modify the dataset by adding a small fraction $\pi$ of random samples from the target to the source for two reasons: (i) our exponential tilt model requires some amount of overlap between source and target; (ii) arguably, in practice, it is more likely that the source dataset has at least a small representation of all groups.

Our goal is to show that ExTRA can identify the target samples mixed into the source for efficient fine-tuning. We obtain feature representations from a pre-trained self-supervised SwAV (Caron et al., 2020). To obtain the ExTRA weights we use SwAV features as sufficient statistic. We then train logistic regression models on (i) the source dataset re-weighted with ExTRA, (ii) uniformly weighted source (S -> T), (iii) target samples mixed into the source ($\pi$T -> T), (iv) all target samples (oracle). See Appendix B.1, B.2 for details. We report performance for varying mixing proportion $\pi$ in Figure 3. First, we note that even when $\pi = 0$, i.e. source and target have completely dis-

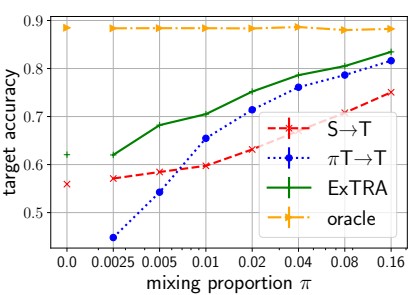

Figure 3: Performance on BREEDS.

joint groups (similar to domain generalization), ExTRA improves over the vanilla S -> T. Next, we see that S -> T improves very slowly in comparison to ExTRA as we increase the mixing proportion; $\pi$T -> T improves faster as we increase the number of target samples it has access to, but never suppresses ExTRA and matches its improvement slope for the larger $\pi$ values. We conclude that ExTRA can effectively identify target samples mixed into source that are crucial for the success of fine-tuning *and* find source samples most relevant to the target task allowing it to outperform $\pi$T -> T. We report analogous precision and recall for the WATERBIRDS experiment in Appendix B.3.

## 7  CONCLUSION

In this paper, we developed an importance weighing method for approximating expectations of interest on new domains leveraging unlabeled samples (in addition to a labeled dataset from the source domain). We demonstrated the applicability of our method on downstream tasks such as model evaluation/selection and fine-tuning both theoretically and empirically. Unlike other importance weighing methods that only allow covariate shift between the source and target domains, we permit concept drift between the source and target. Though we demonstrate the efficacy of our method in synthetic setup of concept drift (Appendix C), in a future research it would be interesting to investigate the performance in more realistic setups (*e.g.* CIFAR10.2 to CIFAR10.2 (Lu et al., 2020), Imagenet to Imagenetv2 (Recht et al., 2019)).

Despite its benefits, the exponential tilt model does suffer from a few limitations. Implicit in the exponential tilt assumption is that the supports of the target class conditionals have some overlap with the corresponding source class conditionals. Although this assumption is likely satisfied in many instances of domain generalization problems (and is always satisfied in the subpopulation shift setting), an interesting avenue for future studies is to accommodate support alignment in the distribution shift model, *i.e.* to align the supports for class conditioned feature distributions in source and target domains. One way to approach this is to utilize distribution matching techniques from domain adaptation literature (Ganin et al., 2016; Sun & Saenko, 2016; Shen et al., 2018), similarly to Cai et al. (2021). We hope aligning supports via distribution matching will allow our method to succeed on domain generalization problems where the support overlap assumption is violated.

## 8 ETHICS STATEMENT

We recommend considering the representation of the minority groups when applying ExTRA in the context of fairness-sensitive applications. The goal of ExTRA is to approximate the distribution of the target domain, thus, in order to use ExTRA weights for fine-tuning or model selection to obtain a fair model, the target domain should be well representative of both privileged and unprivileged groups. If the target domain has miss/under-represented groups, a model obtained using ExTRA weights may be biased.

### ACKNOWLEDGMENTS

This paper is based upon work supported by the National Science Foundation (NSF) under grants no. 2027737 and 2113373.

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
