# OpenReview forum: "Understanding new tasks through the lens of training data via exponential tilting"
_ICLR.cc/2023/Conference — ICLR 2023 poster_

### Official Review · Reviewer_AxBN · 2022-10-23

**Confidence:** 3
**Correctness:** 4
**Technical Novelty And Significance:** 3
**Empirical Novelty And Significance:** 3
**Recommendation:** 6

**Clarity, Quality, Novelty And Reproducibility:**

The work is clearly described. Although importance weights are common in domain shift problems, the paper presents an interesting way to use them that permits novel use cases.

**Strength And Weaknesses:**

Strengths
- The exponential tilt model is well described and the procedure for obtaining the importance weights is clear
- The weights demonstrate value in both domain adaptation and evaluation
- I did not thoroughly check the theoretical proofs but the general steps seem correct

Weaknesses
- Since the method is supposed to work also on domain adaptation, it would be nice to have presented results on some more standard domain adaptation tasks and baselines (e.g., Office).

**Summary Of The Paper:**

This paper introduces a framework for computing importance weights for labeled source data with respect to a target domain where only unlabeled target data is available. Unlike previous methods, these weights allow for concept drift in addition to covariate shift. The paper proves the statistical consistency of the estimators and generalization bounds and demonstrates the efficacy of the methods over two data sets.

**Summary Of The Review:**

Overall, the paper seems strong albeit with limited experiments vs baselines. The theoretical results are nice and the insights are well-supported.

---

> ### Author Response · Authors · 2022-11-16
> **Response to reviewer AxBN**
>
> We thank the reviewer for the comments. We address your concern below.
>
> **Domain adaptation applications:** The domain adaptation settings (e.g., Office) typically consider training a model that is specialized to a combination of domains.
> In our experiments, we considered a fine-tuning setting, where one has already trained a sophisticated model and wants to use it for out-of-distribution prediction without expensive retraining on the full model. For our Waterbirds (Section 5) and Breeds (Section 6) applications, we considered two pre-trained feature models (ResNet18 for Waterbirds and self-supervised SwAV for Breeds) which were later turned into OOD prediction models by training simple target domain-specific heads (logistic regression models) on top of those feature models.  Application of our method to the domain adaptation settings is outside of the focus of our paper. However, we agree that it is an interesting question to consider in a future research.

---

### Official Review · Reviewer_3m7n · 2022-10-24

**Confidence:** 3
**Correctness:** 3
**Technical Novelty And Significance:** 4
**Empirical Novelty And Significance:** 3
**Recommendation:** 6

**Clarity, Quality, Novelty And Reproducibility:**

Clarity: mostly good, a couple of small confusions
Quality: I think the technical work here is pretty good (although not an expert)
novelty: seems like a novel approach to a domain generalization-type problem without group information
Reproducibility: details provided are pretty good, seems reproducible

**Strength And Weaknesses:**

This seems like technically strong work, and the proofs look sound and interesting. The experimental work supports the theoretical components and shows that this method is an interesting possibility for domain adaptation/generalization without group labels.

Some feedback:
- the authors argue that their method moves beyond covariate shift to concept drift - however, the experimental datasets (Waterbirds + Breeds), do not provide any instances of concept drift, only testing covariate shift. Experiments that test the method in a concept drift setting would make the paper's argument stronger
- Prop 4.2: I don't understand what the "T(S_k) is p-dimensional" portion of this assumption means - I think if I understood this piece, then the identifiability argument + concept drift application might be clearer to me. Also, in the appendix, it seems to say d-dimensional instead
-Assumption 4.5 - I don't totally understand this - I've seen constructions like it before but I don't recall how the sequence r_np is used in this statement and it's not clear from the body of the text: would be good to have this be explicit somewhere.
- Experiments: I don't understand why the method "\pi T -> T" doesn't perform close to perfectly on some of these target domain groups. In Waterbirds, some pairs of domains have all the same label (I think for instance environment 0 and 1 both are all Y=0). Then, shouldn't finetuning on data from those domains yield a really easy fine-tuning problem - learning a constant output?

- page 3, choosing T: the intuition provided her around how T should identify "subpopulations" isn't quite clear to me, would appreciate a bit more of a walkthrough of this
- Thm 4.6 - where is M used here? I don't see it but might be missing it
- Lemma 4.8: remind us what G is here, I don't remember

-middle of p7: guaranty -> guarantee



**Summary Of The Paper:**

In this paper, the authors propose a method for domain adaptation when you have labelled data from source P and unlabelled data from target Q. Their method learns a reweighting of the source data sample through a learned exponential tilting, where a neural network learns the sufficient statistics of the tilting. They show that this tilting is identifiable when "anchor sets" exist (a subset of points which only have probability mass for their class label) in the source distribution, and show that when identifiable, the parameter estimate converges. Experimentally, they show that this method is competitive on domain generalization benchmarks.

**Summary Of The Review:**

I think the technical work here provides a clean, effective direction to building approaches for domain adaptation/generalization without group labels. My main question/confusion with the work is re: its connection to concept drift, and I'm not convinced by the paper that this approach is relevant. But even without that, I think the work is good, so I'm recommending acceptance.

---

> ### Author Response · Authors · 2022-11-16
> **Response to reviewer 3m7n**
>
> We thank the reviewer for the detailed feedback and comments. We address questions and concerns below.
>
> **Examples of concept drift:** We have an additional synthetic experiment in Appendix C that is an instance of concept drift and is not covered by subpopulation shift. In our main text, we have referred to that experiment in the last line of the Introduction section and on the third page (the last line in the paragraph named "The exponential tilt model"). In the BREEDS experiment (in Figure 3) we notice that even when no sample is mixed from target to source (i.e. not a subpopulation shift case), ExTRA reweighing does improve model accuracy on the target data when compared to unweighted source data.
>
> **Clarifications:** Below we comment on your clarifying questions.
> - **Prop 4.2:** "$T(S_k)$ is $p$-dimensional" means the image set $T(S_k) = \\{T(x): x \in S_k\\} \subset \mathbf{R}^p$ is not a set of dimension lower than $p$, i.e., contains a $p$-dimensional open ball. The "$T(S_k)$ is $d$-dimensional" in Appendix A.3 is a typo and should be "$T(S_k)$ is $p$-dimensional". Thank you for pointing that out and we have fixed that in our revised draft.
> - **Assumption 4.5:** Fitting our ExTRA weights requires a probabilistic classifier $\hat \eta_{P, k}$ for the source domain (Lemma 3.1) and the quality of our ExTRA weights depends on the quality of that classifier. To observe this phenomenon we assume (in Assumption 4.5) that the logit of the fitted classifier $\hat f_P$ converges to the logit of the true classifier $f_P^\star$ at $r_{n_P}$ rate and see that $r_{n_P}$ appear in the rate of convergence for the ExTRA weights (Theorem 4.6). Therefore, while studying the rate of convergence for the ExTRA weights, Assumption 4.5 is required for controlling the $\ell_2$-error rate for the logit of the source classifier.
> - **In Waterbirds, the same labels for some domain pairs:** All of our chosen group pairs have images of land and waterbirds (0 and 1 as labels) in them. For example, we did not consider the pair {0, 1} that only has images of land birds.
> - **$\pi T\to T$:** In our Waterbirds case study (Section 6)
> $\pi T \to T$ is trained only on those sample points from the source domain that corresponds to a particular target domain. Some of the sample points from other domains could also be similar to those sample points and could be useful in training an out-of-distribution model. Our method can up-weight those samples, thus outperforming $\pi T->T$.
> - **Choosing T:** For a subpopulation shift model where subpopulation identities are known, the $T$ can be set at the subpopulation identities. However, finding such a $T$ can be challenging in cases that are not subpopulation shifts or when the subpopulation identities are not available for a subpopulation shift. In our Waterbirds and Breeds case studies we set $T$ to a simple choice of $T(x) = x$ and find that the choice shows reasonable performance. In a simulation experiment with a normal mixture dataset (in Appendix C) we test the efficacy of such a simple choice ($T(x) = x$) and find that the choice works well for a reasonable range of distribution shifts.
> - **Dependence of $M$ in Thm 4.6:** The constants $k_1$ and $k_2$ depend on $M$ (as appears in Assumption 4.4), in the sense that $k_1$ and $k_2$ increase as $M$ increases. The explicit dependence is quite complicated and is hidden for a simple exposition of the results.
> - **$G$ in Lemma 4.8:** Here, $\mathcal{G} = \\{ \omega^\star(x, y) \ell(f(x), y): \ f \in \mathcal{F}\\}$. We have added the definition of $\mathcal{G}$ in Lemma 4.8 in our revision.

---

> > ### Comment · Reviewer_3m7n · 2022-11-17
> > **Response**
> >
> > Thanks for your clarifications.
> >
> > Concept drift experiments: I don't see these in the revision, am I missing it? should I be able to see it yet?
> >
> > Assumption 4.5: I guess this is standard notation - it's been a while since I worked with this stuff. My main confusion was around how the sequence r_np is indexed - I guess it's indexed by n?

---

> > > ### Author Response · Authors · 2022-11-17
> > > **Response to reviewer 3m7n**
> > >
> > > Thank you for getting back. Please see our responses below.
> > >
> > > **Concept drift experiments:** That was a mistake from our part and thank you very much for pointing that out. We have updated the revised draft that includes the supplementary details and you should be able to find the concept drift example in Appendix C.
> > >
> > > **Assumption 4.5** Yes, you're right. The sequence $r_{n_P}$ is indexed by $n_P$, the sample size in the source domain.

---

### Official Review · Reviewer_UqFX · 2022-10-25

**Confidence:** 4
**Clarity, Quality, Novelty And Reproducibility:** See strengths and weaknesses above
**Correctness:** 4
**Technical Novelty And Significance:** 3
**Empirical Novelty And Significance:** 3
**Recommendation:** 6

**Strength And Weaknesses:**

Strengths

- The paper is well-written. The exponential tilt model is clearly explained.
- Analyzing learned importance weights.


Weaknesses
- Exponential tilt assumption unjustified. It is not clear why complex distribution shits in practice should be parameterized with the exponential tilt model. The paper states that this problem can be applied to concept drift settings but has no experiments / analysis on concept drift.

- When does this re-weighting approach fail? A more quantitative approach to this question would be insightful. The paper loosely talks about this  (i.e. distribution needs some overlap etc).

- Limited empirical evaluation. Waterbirds and Entity30 are datasets where you know what the "ground-truth" importance weights should look like (https://proceedings.mlr.press/v177/idrissi22a.html). Showing that the proposed methods works well in these settings is a good first paper, i think the paper will be significantly stronger if the paper improves performance on more general / realistic distribution shifts (e.g. CIFAR 10.2, ImagenetV2) and then analyzes the learned importance weights. The empirical section should have additional baselines (e.g. https://proceedings.mlr.press/v162/zhou22d/zhou22d.pdf, https://proceedings.mlr.press/v177/idrissi22a.html) to clearly contrast this approach from previous methods.

- Theoretical analysis somewhat tangential and not insightful vis-a-vis the paper's main focus. I would rather first read whether this method works well on realistic distribution shifts and then discuss properties like consistency and identifiability of the parameters.

Overall the paper is well-written and focuses on a principled approach (reweighting) to improve OOD performance. However, the paper has two major issues: (a) exponential tilt assumption is not clearly justified, (b) empirical evaluation is quite limited.


**Summary Of The Paper:**

This paper considers the problem of reweighting training samples to improve model performance on out-of-distribution test samples. The approach formulates real distribution shifts (covariate and concept related) using the exponential tilt assumption. With this assumption, the problem of improving performance on OOD samples simplifies to learning data importance weights. The paper has some theoretical analysis on the properties of exponential tilting. The paper also applies this method to improve performance on Waterbirds and BREEDS-Entity30.

**Summary Of The Review:**

See strengths and weaknesses above

---

> ### Author Response · Authors · 2022-11-16
> **Response to reviewer UqFX**
>
> We thank the reviewer for the comments. We address questions and concerns below.
>
> **Justification of exponential tilt assumption:** The exponential tilt model is quite flexible. For any smooth source and target class conditionals, it is possible to have a sufficient statistic that approximates the test distribution arbitrarily well. Though the suitable sufficient statistic may be unknown to the learner, weighted source data with learned weights from the exponential tilt model would still better approximate the target distribution than uniformly weighted source data.
>
> **Applications to concept drift outside of subpopulation shift:** We have an additional synthetic experiment in Appendix C where we have considered an instance of concept drift that is not covered by subpopulation shift. In our main text, we have referred to that experiment in the last line of the Introduction section and on the third page (the last line in the paragraph named "The exponential tilt model"). In the BREEDS experiment (in Figure 3) we notice that even when no sample is mixed from target to source (i.e. not a subpopulation shift case), ExTRA reweighing does improve model accuracy on the test data when compared to unweighted training data.
>
> **Failure cases of reweighting:** As we state in the paragraph entitled "The exponential tilt model" on the third page and in the second paragraph of the conclusion section, the reweighting approach may fail if there is no overlap between the source and target domains. To this effect, we have performed a quantitative investigation in the synthetic experiment on a normal mixture model in Appendix C, where the overlap between the source and target distributions is smoothly controlled by a parameter $\delta$. We observe that the quality (Wasserstein distance between ExTRA weighted source and uniformly weighted target distributions, as observed in Figure 10) and performance (target accuracy in Figure 9) of our ExTRA weights are good on a reasonable range of $\delta$, meaning a reasonable overlap between the domains. However, as to be expected, the ExTRA weights have poor quality and performance for large $\delta$, when the overlap is very little.
>
> **Known "ground-truth" importance weights in Waterbirds and Entity30 datasets:** Though the "ground-truth" about the importance weights is known for Waterbirds and Entity30 datasets through the group annotations, one of the advantages of our method is that we did not require these group annotations for estimating the ExTRA weights. We have only used the group annotations for evaluating the quality of our weights via precision-recall curves, in addition to evaluating OOD performance for these weights. Note that these are standard benchmarks for evaluating OOD performance for sample reweighting methods [3].
>
>
> It would be interesting to examine the performance of our ExTRA weights in the datasets that you're referring to and we shall try to do that in a time-permitting manner, but for now, we have mentioned applications to these datasets as potential future works in our conclusion section.
>
> **Additional baselines:** The reweighting methods in [2] are aimed to address the issue of out-of-distribution generalization in over-parameterization scenarios. On the contrary, we use the reweighting method to fine-tune a pre-fitted model to a particular target domain. Most of the methods considered in [1], such as ERM, group distributionally robust optimization (gDRO), and group reweighting have already been considered in our paper. The just-train-twice (JTT) benchmark, which weights the source samples with respect to their model errors, uses the same principle as a distributionally robust optimization (DRO), which we have considered in our experiments.
>
> Moreover, our baselines in Waterbirds experiments are quite comprehensive: (1) group reweighting and gDRO, which uses group attributes, (2) DRO, which does not use group attributes, (3) $\pi T\to T$, which uses oracle weights on source samples, and (4) $S\to T$, the simple ERM model.
>
> ---
> ## References:
>
> [1] Idrissi, Badr Youbi, et al. "Simple data balancing achieves competitive worst-group-accuracy." Conference on Causal Learning and Reasoning. PMLR, 2022.
>
> [2] Zhou, Xiao, et al. "Model agnostic sample reweighting for out-of-distribution learning." International Conference on Machine Learning. PMLR, 2022.
>
> [3] Garg, Saurabh, et al. "Leveraging unlabeled data to predict out-of-distribution performance." arXiv preprint arXiv:2201.04234 (2022).

---

> > ### Author Response · Authors · 2022-11-16
> > **Response to reviewer UqFX**
> >
> > **Insights for the theoretical results:** Most of the works in the literature related to sample reweighting lack theoretical justifications for their proposed methods. In our work, we provide a statistically principled approach to sample reweighting, whose required assumptions and consistency results are provided in Sections 3 and 4.
> >
> > The required assumptions can provide some insights into the limitations of our reweighting setup. For example, an implication of our exponential tilt assumption (stated in equation (3.2)) is that our reweighting approach will perform poorly when there is very little to no support overlap between the source and target class conditional distributions. In Appendix C we have investigated this in a synthetic experimental setup. Comparing Assumption 4.5 and Theorem 4.6 one can see that the quality of our ExTRA weights depends on the quality of the probabilistic classifier used for estimating these weights. Under those assumptions, we study the qualities for the downstream tasks, such as (1) the target performance of a model fitted with ExTRA weighted source data (in Lemma 4.8) and (2) the ability to evaluate target performance (in the paragraph named "Target performance evaluation") in terms of the quality of our ExTRA weights.

---

> > > ### Comment · Reviewer_UqFX · 2022-11-19
> > > **Post-rebuttal response**
> > >
> > > Thank you, the rebuttal addresses my main concerns and questions, so I am raising my score to 6. However, as I mentioned in the review, the paper would (in my opinion) greatly benefit from restructuring and more thorough empirical evaluation.

---

### Decision · Program_Chairs · 2023-01-20

**Decision:**

Accept: poster

**Justification For Why Not Higher Score:**

There are some reservations from the reviewers.

**Justification For Why Not Lower Score:**

Despite some reservations, the paper proposes a well-supported method for dealing with the distribution shift problem.

**Metareview: Summary, Strengths And Weaknesses:**

This paper investigates the distribution shift problem in which train and test distributions can differ and one has access to the unlabeled data from the test domain. The paper adopts a reweighting approach, which is quite common in the literature. The goal is to learn a weight function w(X,Y) such that the moment conditions

E[w(X_P,Y_P)g(X_P,Y_P)] = E[g(X_Q,Y_Q)]

are fulfilled for all real-valued function g(X,Y). Motivated by the rich theory of exponential families in statistics, the authors propose a parametric model of the weight function, called the exponential tilt model

w(x,y) = exp(a_y.T(x) + b_y)

where T(x) is a vector of sufficient statistics and a_y,b_y are the parameters. The paper proposes ways to construct T(X), to learn the parameters a_y,b_y as well as the corresponding weights. The paper also provides theoretical analyses of the proposed method. The empirical results on two standard benchmark datasets are promising.

Overall, there is a general consensus among the expert reviewers that this is a strong paper despite some reservations. The aforementioned moment conditions are well-known and the reweighting approach is standard in the distribution shift literature. What makes this paper interesting however is the introduction of a "parametric exponential tilt model" on the weight function, allowing the in-depth theoretical analysis and capability of the model to deal with the "concept drift" scenario. Note that this concept drift scenario would have been much more challenging in a nonparametric setting. I find this aspect of the paper quite interesting and would be beneficial for other researchers working in this area.

Some caveats of the paper include weak justification of the exponential tilt model and limited experiments/baselines. I would strongly encourage the authors to improve these two limitations in the camera-ready version.

As a result, I recommend this paper with expected minor improvements in the camera-ready for publication at ICLR 2023.

**Note From Pc:**

if the above contains the word "oral" or "spotlight" please see: "oral" presentation means -> notable-top-5% and "spotlight" means -> notable-top-25%. As stated in our emails, we are disassociating presentation type from AC recommendations

**Summary Of Ac-Reviewer Meeting:**

N/A